# Minigene-Based Splice Assays Reveal the Effect of Non-Canonical Splice Site Variants in *USH2A*

**DOI:** 10.3390/ijms232113343

**Published:** 2022-11-01

**Authors:** Janine Reurink, Jaap Oostrik, Marco Aben, Mariana Guimarães Ramos, Emma van Berkel, Monika Ołdak, Erwin van Wijk, Hannie Kremer, Susanne Roosing, Frans P. M. Cremers

**Affiliations:** 1Department of Human Genetics, Radboud University Medical Center, 6525 GA Nijmegen, The Netherlands; 2Donders Institute for Brain Cognition and Behaviour, Radboud University Medical Center, 6525 AJ Nijmegen, The Netherlands; 3Department of Otorhinolaryngology, Radboud University Medical Center, 6525 GA Nijmegen, The Netherlands; 4Department of Genetics, Institute of Physiology and Pathology of Hearing, 02-042 Warsaw, Poland

**Keywords:** *USH2A*, Usher syndrome, retinitis pigmentosa, non-canonical splice site, minigene splice assay, pre-mRNA splicing

## Abstract

Non-canonical splice site variants are increasingly recognized as a relevant cause of the *USH2A*-associated diseases, non-syndromic autosomal recessive retinitis pigmentosa and Usher syndrome type 2. Many non-canonical splice site variants have been reported in public databases, but an effect on pre-mRNA splicing has only been functionally verified for a subset of these variants. In this study, we aimed to extend the knowledge regarding splicing events by assessing a selected set of *USH2A* non-canonical splice site variants and to study their potential pathogenicity. Eleven non-canonical splice site variants were selected based on four splice prediction tools. Ten different *USH2A* constructs were generated and minigene splice assays were performed in HEK293T cells. An effect on pre-mRNA splicing was observed for all 11 variants. Various events, such as exon skipping, dual exon skipping and partial exon skipping were observed and eight of the tested variants had a full effect on splicing as no conventionally spliced mRNA was detected. We demonstrated that non-canonical splice site variants in *USH2A* are an important contributor to the genetic etiology of the associated disorders. This type of variant generally should not be neglected in genetic screening, both in *USH2A-*associated disease as well as other hereditary disorders. In addition, cases with these specific variants may now receive a conclusive genetic diagnosis.

## 1. Introduction

Aberrant splicing has been increasingly recognized as an important mechanism in human disease. It is currently estimated that up to a third of variants that are involved in disease have an effect on pre-mRNA splicing, including variants in trans-regulatory factors that bind to pre-mRNA as well as variants within the pre-mRNA itself [1].

An effect on pre-mRNA splicing has been experimentally confirmed for several variants in the *USH2A* gene, which is associated with autosomal recessive retinitis pigmentosa (arRP) and Usher syndrome type 2 (USH2), including pathogenic deep-intronic variants [2,3,4,5]. In addition to these deep-intronic variants, ~150 and ~330 *USH2A* non-canonical splice site (NCSS) variants have been reported in the Leiden Open-source Variation Database (LOVD) [6] and ClinVar [7], respectively (accessed in June 2022). To the best of our knowledge, for 13 of these variants a splice defect has been confirmed using different methods, such as minigene splice assays or the analysis of RNA derived from nasal epithelial cells [8,9,10,11,12]. The remainder of these variants is either classified as variant of uncertain significance (VUS) or their classification relies on in silico prediction tools only.

The description of numerous NCSS variants in LOVD and ClinVar indicates that variants of this type have been identified regularly in cases with *USH2A-*associated disease. Functional validation studies will benefit classification of these variants and thereby increase the number of cases that receive a conclusive genetic diagnosis. There are several upcoming exon- and mutation-specific therapies for *USH2A,* such as splice modulation [13], CRISPR/Cas9-based editing [14] and exon excision [15], as well as for other genes associated with inherited retinal disease (IRD) [16,17]. Cases with a conclusive genetic diagnosis could now also become eligible for future therapy.

In this study, we selected a set of *USH2A* NCSS variants that have not been functionally validated previously and demonstrated their effect on pre-mRNA splicing in minigene splice assays with the aim of assessing their pathogenicity and to increase our understanding of the effect of this type of variants in human disease in general.

## 2. Results

### 2.1. Eleven Variants were Selected for Minigene Splice Assays

We assessed NCSS variants that were reported to be *in trans* with another (likely) pathogenic *USH2A* variant in the *USH2A* LOVD (accessed in August 2019) [6] for the average decrease in splice site strength predicted by four splice prediction tools in Alamut Visual Plus (v1.4); SpliceSiteFinder-like [18], MaxEntScan [19], NNSPLICE [20] and GeneSplicer [21]. Eleven variants with a predicted average decrease ≥10% were selected for minigene splice assays (Table 1). Figure 1 visualizes their location within the *USH2A* gene and the minigene constructs that were generated. Two of the selected variants were predicted to affect the canonical splice acceptor site, while the majority of selected variants (*n* = 9) were predicted to have an effect on the canonical splice donor site. The average predicted decrease in splice site strength, based on the four prediction programs, ranged from 13.3% (for c.2994−10T>G) to 84.4% (c.15519+2dup).

### 2.2. Assessment of Pre-mRNA Splicing Using Minigene Splice Assays

Ten minigene constructs were generated to assess the putative effect on pre-mRNA splicing of the 11 selected *USH2A* variants. All variants had an effect on pre-mRNA splicing in our minigene splice assays (Figure 2), with skipping of one entire exon being the most frequently observed effect. Eight variants (73%) induced non-canonical splicing events in all detected transcripts, as we did not detect conventionally spliced mRNA upon expression of the mutant constructs.

Four of the evaluated variants induced a frameshift in all detected transcripts. Variant c.5168G>A resulted in skipping of exon 26 (−131 nucleotides (nt)) and dual skipping of exons 25 and 26 (−311 nt), resulting in premature termination in all transcripts (p.[Glu1663Aspfs*31,Gly1723Aspfs*31], Figure 2B). Variants c.10182G>A and c.14791+4A>G resulted in skipping of respectively exon 51 (−205 nt, p.Met3321Asnfs*22, Figure 2F) or exon 67 (−209 nt, p.Tyr4862Alafs*22, Figure 2I). Variant c.10939G>A (p.[Val3582Serfs*26,Gln3627Serfs*35], Figure 2G) induced a dual out-of-frame effect; we observed both a deletion of the entire exon 55 (−199 nt) as well as a very faint band representing the deletion of part of exon 55 (−62 nt).

Four variants resulted in a(n) (partial) in-frame skipping event. Variant c.5167G>C resulted in the in-frame skipping of exon 25 (−180 nt) and out of frame co-skipping of exons 25 and 26 (−311 nt), resulting in p.[Glu1663Aspfs*31,Glu1663_Ala1722del] (Figure 2B). Three variants resulted in an in-frame exon skipping event in all detected transcripts: c.5776G>A (exon 28, −204 nt, p.Gly1858_Thr1925del, Figure 2C), c.8681G>A (exon 43, −123 nt, p.Tyr2854_Arg2894del, Figure 2D) and c.15519+2dup (exon 71, −222 nt; p.Gly5100_Leu5173del, Figure 2J).

The remaining three of the 11 variants (c.2994−10T>G, c.9958G>T and c.11389+3A>G) only showed an incomplete effect on pre-mRNA splicing, resulting in the (partial) skipping of exon 15 (−164 nt, p.[=,Arg998Serfs*26], Figure 2A), exon 50 (−61 nt and −219 nt, p.[Gly3320Cys,Ser3301Cysfs*9,Glu3248_Gly3320del], Figure 2E) or exon 58 (−43 nt and −158 nt, p.[Tyr3745Aspfs*16,Val3783Glyfs*20], Figure 2H), respectively. All results are summarized in Table 2.

## 3. Discussion

All 11 tested variants had an effect on pre-mRNA splicing in our minigene splice assays, with skipping of an entire exon being (one of) the effect(s) for all variants. Eight of the variants showed a complete effect on splicing, whereas for three variants remaining conventionally spliced transcript was detected. Our results confirm that NCSS variants, including missense variants and synonymous variants, are an important contributor to pathogenicity in *USH2A*-associated disease, as we observed an effect on splicing for 100% of the variants with no remaining conventionally splices transcript for 73% of the variants.

The solve rate observed in studies such as ours are highly dependent on the inclusion criteria that are employed, including the tools that were employed for selection of variants and how stringent the inclusion thresholds were that were used. Our results are in line with previous functional studies on NCSS variants in genes associated with IRD. Previous studies observed effect ratios ranging from 76% (16/21) of NCSS variants in genes associated with Usher syndrome [9], 94% (44/47) of variants in an *ABCA4-*specific study [30] and four out of four NCSS variants in a study focusing on variants associated with IRD in general [31].

Interestingly, the majority (9/11; 82%) of variants that were included in our study have an effect on the splice donor site, including six variants that were affecting a guanine at the last nucleotide of an exon. Only two of the variants were predicted to affect the splice acceptor site. This imbalance has been observed previously in a review discussing *ABCA4* variants, in which 69% (44/64) of non-canonical splice site variants affected the canonical splice donor site [32]. It has been suggested that this imbalance is a result of a longer consensus sequence of the splice acceptor site, with less effect of variants affecting the pyrimidine stretch unless a new splice acceptor site is created [30]. However, a study on NCSS variants detected in genes associated with Usher syndrome did not find a disbalance between variants as they confirmed an effect on splicing for nine out of eleven variants near the splice donor site and seven out of ten variants adjacent to the splice acceptor site that were detected [9].

Our selection of NCSS variants is based on four splice prediction tools embedded in Alamut Visual, that predicted an average decrease in canonical splice site strength ranging from 13.3 to 84.4%, with a median of 48%. Three variants for which an effect on splicing was observed for only a part of the transcripts (c.2994−10T>G, c.9958G>T and c.11389+3A>G) had a predicted reduction in splice site strength of 13.3%, 27.8% and 18.6%, respectively. For c.9958G>T, the predicted decrease was among the lowest, but not among the three lowest, average prediction score. In contrast, the predicted decrease in splice site strength was less strong for another variant (c.5168G>A) for which a complete effect on splicing was observed on all transcripts detected in our minigene splice assay. This highlights that NCSS variants should be experimentally verified, and that their potential effects should not solely be extracted based on splice prediction tools. The average predicted decrease of >10% of four in silico prediction tools (SpliceSiteFinder-like [18], MaxEntScan [19], NNSPLICE [20] and GeneSplicer [21]) that was used as inclusion criterium in our study is likely too stringent. A lower inclusion threshold may yield more variants with a(n) (partial) effect on splicing in future studies.

In recent years, several splice prediction tools have become available, with different strengths and weaknesses, including machine-learning and deep-learning tools [33]. We compared splice prediction scores from the tools in Alamut Visual to scores obtained from a deep-learning-based prediction program (SpliceAI [34]) (Appendix A, Appendix A), to determine differences in splice prediction accuracy. A weak association between scores produced by both tools is visible and the three variants with remaining conventionally spliced RNA have among the lowest scores for both tools, but not the three absolute lowest scores, demonstrating again that functional evidence should be collected for variants with a predicted effect on splicing. A study by Rowlands et al. tested the capability of eight in silico splice prediction tools to predict an effect on pre-mRNA splicing of 249 functionally verified VUS. SpliceAI was the best performing tool in their study and they calculated that the optimal threshold for inclusion of variants is 0.145 (for the highest SpliceAI delta score) [35]. The highest SpliceAI predicted delta scores were well above 0.145 for all 11 tested variants in our study and all variants were shown to have an effect on pre-mRNA splicing. Our results for *USH2A* are therefore in line with the findings of Rowlands et al..

For two of the variants (c.5167G>C and c.5168G>A) we observed dual exon skipping of exons 25 and 26. This could be an artefact of the assay that was potentially enhanced by the presence of the variants, as a weak band at the same height was also observed for the wildtype construct (Figure 2B). The same applies to c.10182G>A as a similar weak band was observed upon transfection of the wildtype construct that could indicate exon skipping (Figure 2F). In addition, we observed inclusion of a 39-nt pseudo-exon in intron 15 (c.3157+84_3157+122) in wildtype (WT) and mutant (M) mRNA during assessment of c.2994−10T>G (Figure 2A). With the inclusion of at least 1000 nt upstream and downstream of the canonical splice sites, we aimed to prevent potential artefacts, but with the limitations of an assay containing only part of the *USH2A* gene, such observations are inevitable. Use of larger minigenes or testing variants in patient-derived cells expressing native *USH2A* could elucidate to what extent our observations of dual exon skipping and pseudo-exon inclusion are artefacts of the assay or true events. In line with these observations, it is possible that dual exon skipping is also caused by other variants in our study and was not observed due to construct size limitations. Ideally, one should test all variants in a cellular system that better resembles the in vivo situation, such as photoreceptor precursor cells or organoids. These models better mimic the human retina with expression of native, full-length *USH2A*, including other variants *in cis* that may influence the observed effect on splicing. Use of such models could result in more accurate observations, as it is known that splice patterns in retina and inner ear can deviate from those in other cells or tissues [36,37]. However, as we selected the variants from literature, we did not have access to material from the cases with the variants of interest and a minigene splice assay was the most feasible method to observe any effect on splicing in vitro.

In conclusion, our study demonstrates that the NCSS variants in *USH2A* that we selected all result in aberrant splicing. This type of variation is shown to be an important contributor to pathogenicity and should not be neglected in genetic screening in *USH2A-*associated disease as well as other hereditary disorders. Our findings will contribute to providing a genetic diagnosis to cases with these variants and their potential eligibility for future therapy.

## 4. Materials and Methods

### 4.1. Variant Selection

All variants in the *USH2A* gene (*n* = 1499, NM_206933.2) were retrieved from LOVD and we excluded variants that were; (1) only reported by ‘Vereniging klinisch genetische laboratoriumdiagnostiek’, as the second variant was not reported for these cases; (2) frameshift or stop gain variants, because these are expected to be pathogenic regardless of the effect on splicing; (3) already classified as (likely) benign; (4) previously shown to have an effect on pre-mRNA splicing in literature; or (5) insertions or deletions larger than 1 nt.

The remaining variants (*n* = 381) were manually prioritized for being in the NCSS region, being intronic variants located 3 to 15 nt upstream of a canonical splice acceptor site or 3 to 6 nt downstream of a canonical splice donor site and exonic variants located at a maximum of 5 nt from a canonical splice site. For the remaining 31 variants, we screened the corresponding publications in which they were reported for the presence of a second *USH2A* variant that was classified as a (likely) pathogenic variant or VUS. For variants that were reported to be in a compound heterozygous configuration with another *USH2A* variant, we assessed the average predicted decrease in splice site strengths, based on four splice prediction tools in Alamut Visual Plus (v1.4); SpliceSiteFinder-like [18], MaxEntScan [19], NNSPLICE [20] and GeneSplicer [21]. Ten variants published by Baux et al. in 2014 [22], Lenassi et al. in 2015 [23], Reddy et al. in 2014 [24], Jiang et al. in 2015 [25], Glöckle et al. in 2013 [26], Krawitz et al. in 2014 [27], Bonnet et al. in 2016 [28] and Koyanagi et al. in 2019 [29], with an average predicted decrease of >10% on the strength of the nearby canonical splice site, were selected for testing. An 11th variant (c.2994−10T>G) was added after personal correspondence.

### 4.2. Minigene Splice Assays

The exon(s) of interest, including at least 1000 nt of the flanking upstream and downstream intronic sequence, were amplified from genomic DNA with PCR according to a standard protocol (Primer sequences are in Appendix A). WT constructs were first cloned into the pDONR™201 vector (Thermo Fisher Scientific, Carlsbad, CA, USA) and then into a vector containing *RHO* exons 3 and 5 (pCI-neo) using Gateway^®^ cloning technology (Thermo Fisher Scientific, Carlsbad, CA, USA) as previously published [38]. If variants were in the vicinity of each other, one construct was generated for both variants. A mutagenesis PCR was performed on pDONR™201 constructs to create a M construct containing the variant of interest from the WT construct, followed by a recombination reaction to transfer the insert into the pCI-neo vector, as published previously [30]. All constructs were verified with Sanger sequencing to exclude the presence of other rare variants (gnomAD allele frequency <1%) that could potentially affect splicing according to the four splice prediction tools in Alamut Visual Plus. One variant (c.5857+769G>T) did not meet these criteria and was removed from the corresponding construct with mutagenesis PCR. WT and M constructs (500 ng) were then individually transfected in HEK293T cells with FuGENE^®^ HD Transfection Reagent (Promega, Madison, WI, USA) and cells were harvested 24 h post-transfection. Total RNA was isolated using the NucleoSpin RNA Clean-up kit (Macherey-Nagel, Düren, Germany) and cDNA was generated from 500 ng of RNA using the iScript cDNA Synthesis kit (Bio-Rad, Hercules, CA, USA) following manufacturer’s instructions. An independent biological replicate transfection was performed for each variant using poly-ethylenimine (100 µg/mL in 150 mM NaCl) as transfection reagent.

PCR was performed on cDNA with Q5 polymerase (New England Biolabs, Ipswich MA, USA) using primers for *RHO* exons 3 and 5 (Appendix A) to assess the effect of all variants on pre-mRNA splicing. To confirm visually observed effects on splicing, PCR products were Sanger sequenced. Variants were classified based on the guidelines of the American College of Medical Genetics and Genomics (ACMG) [39,40].

## Figures and Tables

**Figure 1 ijms-23-13343-f001:**
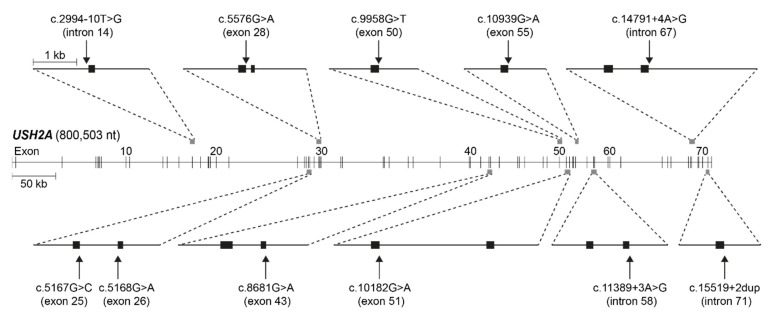
Visual representation of *USH2A* (NM_206933.2) and its exons with the location of the 11 tested variants and ten minigene constructs that were generated. Exons are represented by black blocks. kb: kilobases, nt: nucleotides.

**Figure 2 ijms-23-13343-f002:**
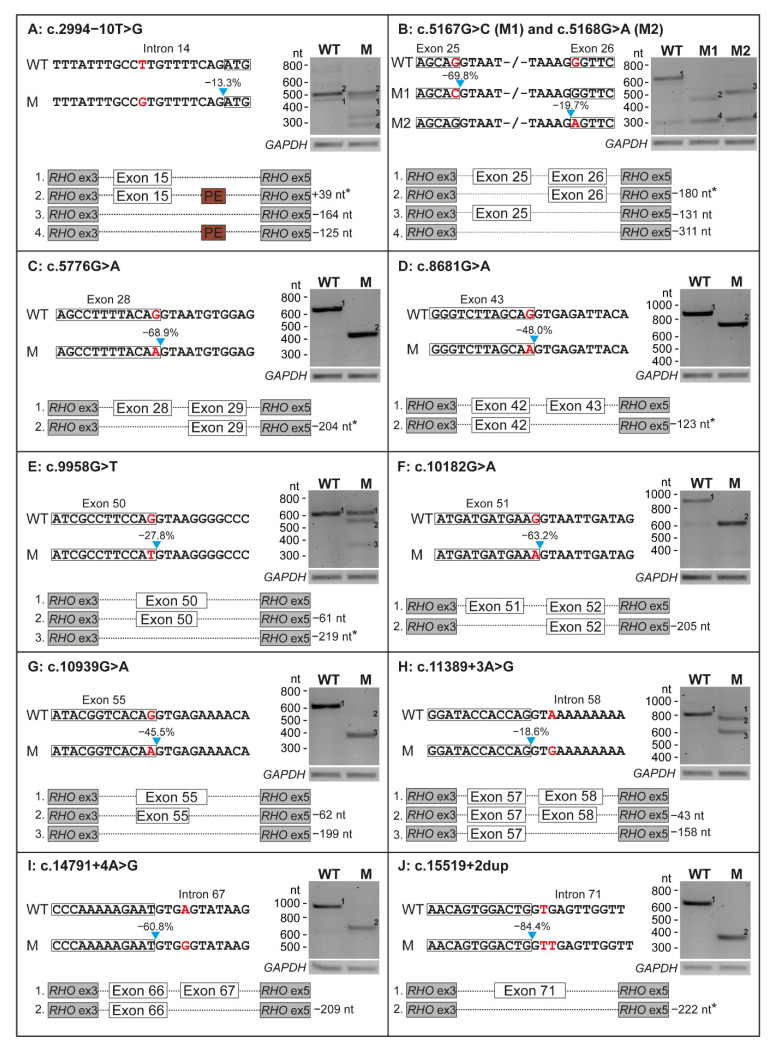
Results on splicing of all 11 variants that were tested in minigene splice assays. For each variant, part of the wildtype (WT) and the mutant (M) construct is depicted. The location of the variant is indicated in red and the predicted decrease in splice site strength is presented above the blue arrow. The results of transfection with both WT and M constructs in HEK293T cells and subsequent RT-PCR and agarose gel electrophoresis are shown for all variants. A schematic overview of all the mRNA products that resulted from transfection with both constructs is visualized. Effects resulting in an in-frame deletion or insertion are marked with an asterisk. *GAPDH* was used as loading control. Ex: exon, nt: nucleotides.

**Table 1 ijms-23-13343-t001:** Non-canonical splice site variants that were selected for a minigene splice assay.

NCSS Variant; cDNA (NM_206933.2)	NCSS Variant; Protein	Variant Location	Pheno Type	Reference	SpliceSite Finder-Like		MaxEnt Scan		NN SPLICE		Gene Splicer		Predicted Average Splice Site Strength Decrease (%)	*USH2A* Variant 2; cDNA, Protein
					WT	M	WT	M	WT	M	WT	M		
	Variants with a predicted effect on canonical splice acceptor sites													
c.2994−10T>G	p.(?)	Intron 14	USH	Personal communication	88.0	82.1	9.5	8.8	1.0	0.8	8.3	6.7	13.3	c.8224del, p.(Val2742Serfs*27)
c.5168G>A	p. (Gly1723Glu)	Exon 26	USH	Baux et al. 2014 [22]	89.1	85.2	7.9	5.8	0.9	0.7	6.2	4.6	19.7	c.908G>A, p.(Arg303His)
	Variants with a predicted effect on canonical splice donor sites													
c.5167G>C	p. (Gly1723Arg)	Exon 25	USH	Baux et al. 2014 [22]	82.6	69.3	9.4	2.3	0.8	0.1	2.4	0.0	69.8	c.2276G>T, p.(Cys759Phe)
c.5776G>A	p. (Glu1926Lys)	Exon 28	arRP	Lenassi et al. 2015 [23]	82.6	70.5	9.4	2.1	0.6	0.1	3.7	0.0	68.9	c.10073G>A, p.(Cys3358Tyr)
c.8681G>A	p. (Arg2894Lys)	Exon 43	USH	Reddy et al. 2014 [24]	90.3	78.2	9.2	5.3	0.6	0.1	0.0	0.0	48.0	c.8681G>A, p.(Arg2894Lys)
c.9958G>T	p. (Gly3320Cys)	Exon 50	USH	Jiang et al. 2015 [25]	95.0	82.4	11.1	6.6	1.0	0.9	8.2	4.3	27.8	c.6488G>A, p.(Trp2163*)
c.10182G>A	p.(Lys3394=)	Exon 51	arRP	Glöckle et al. 2013 [26]	87.3	75.2	8.8	3.6	1.0	0.2	1.5	0.0	63.2	c.10182G>A, p.(Lys3394=)
c.10939G>A	p. (Gly3647Ser)	Exon 55	USH	Krawitz et al. 2014 [27]	90.3	78.2	9.2	5.3	1.0	0.3	8.9	3.9	45.5	c.7139_7140del, p.(Leu2380Profs*37)
c.11389+3A>G	p.(?)	Intron 58	USH	Bonnet et al. 2016 [28]	82.5	78.2	8.6	5.1	1.0	0.9	0.0	0.0	18.6	c.2209C>T, p.(Arg737*)
c.14791+4A>G	p.(?)	Intron 67	USH	Bonnet et al. 2016 [28]	82.7	72.6	7.8	2.8	1.0	0.2	5.4	0.7	60.8	c.10712C>T, p.(Thr3571Met)
c.15519+2dup	p.(?)	Intron 71	arRP	Koyanagi et al. 2019 [29]	87.1	54.4	10.1	0.0	0.9	0.0	7.7	0.0	84.4	c.10859T>C, p.(Ile3620Thr)

All variants are listed, as well as the splice site strength predictions from the four splice prediction tools and the phenotype and the variant that was reported as second *USH2A* variant in each respective case. arRP: autosomal recessive retinitis pigmentosa, M: mutant, NCSS: non-canonical splice site, USH: Usher syndrome, WT: wildtype.

**Table 2 ijms-23-13343-t002:** Protein effects of the splicing aberrations caused by the 11 selected variants.

NCSS Variant (NM_206933.2)	Variant Location	Minigene Splice Assay Effects	Protein Effect	ACMG Classification
c.2994−10T>G	Intron 14	1. Conventionally spliced mRNA, 2. Skipping of exon 15, (3. Inclusion of a 39-nt PE in both WT and M constructs)	p.[=,Arg998Serfs*26]	VUS
c.5167G>C	Exon 25	1. Dual skipping of exons 25 and 26, 2. skipping of exon 25	p.[Glu1663Aspfs*31, Glu1663_Ala1722del]	Likely pathogenic
c.5168G>A	Exon 26	1. Dual skipping of exons 25 and 26, 2. skipping of exon 26	p.[Glu1663Aspfs*31, Gly1723Aspfs*31]	Pathogenic
c.5776G>A	Exon 28	Skipping of exon 28	p.Gly1858_Thr1925del	Likely pathogenic
c.8681G>A	Exon 43	Skipping of exon 43	p.Tyr2854_Arg2894del	Likely pathogenic
c.9958G>T	Exon 50	1. Conventionally spliced mRNA, 2. Skipping of last 61 nt of exon 50, 3. skipping of exon 50	p.[Gly3320Cys,Ser3301Cysfs*9, Glu3248_Gly3320del]	Likely pathogenic
c.10182G>A	Exon 51	Skipping of exon 51	p.Met3321Asnfs*22	Pathogenic
c.10939G>A	Exon 55	1. Skipping of exon 55, 2. skipping of last 62 nt of exon 55	p.[Val3582Serfs*26, Gln3627Serfs*35]	Pathogenic
c.11389+3A>G	Intron 58	1. Skipping of exon 58, 2. skipping of last 43 nt of exon 58, 3. conventionally spliced mRNA,	p.[Tyr3745Aspfs*16, Val3783Glyfs*20,=]	VUS
c.14791+4A>G	Intron 67	Skipping of exon 67	p.Tyr4862Alafs*22	Pathogenic
c.15519+2dup	Intron 71	Skipping of exon 71	p.Gly5100_Leu5173del	VUS

For variant c.9958G>T (p.(Gly3320Cys)), remaining conventionally spliced mRNA was observed in the minigene splice assay and classification according to the guidelines of the American College of Medical Genetics and Genomics (ACMG) was therefore based on the classification of the missense variant. M: mutant, NCSS: non-canonical splice site, VUS: variant of uncertain significance, WT: wildtype.

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
