# Peer review of "Minigene-Based Splice Assays Reveal the Effect of Non-Canonical Splice Site Variants in USH2A"

_ijms, 2022, doi:10.3390/ijms232113343_

Round 1

Reviewer 1 Report

Pathogenic sequence variants in the USH2A gene lead to autosomal recessive retinitis pigmentosa (RP) or Usher syndrome in the majority of cases. Approximately 1,900 variants have been described as disease associated and roughly 10-15% of them are likely involved in correct splicing of the pre-mRNA of the USH2A gene. This number may be higher as non-canonical splice site variants (NCSSVs) might be responsible as well in a significant number of cases.

Reurink et al. report the functional characterization of 11 of such NCSSVs in the USH2A gene, detected in patients with RP or Usher syndrome in combination with a second pathogenic variant in trans. Seven occurred in protein coding parts of the gene, leading to 6 missense and 1 synonymous variant on protein level. In addition, they were predicted as potentially altering splicing. Four of the variants were found in introns, three of which in donor sites and one in an acceptor. Compound heterozygosity with a second pathogenic variant would explain the disease in these cases but interpretation of pathogenicity of NCSSVs is limited by functional analyses.

Eleven NCSSVs were characterized with respect to their impact on splicing by making use of mini gene assays in HEK293T cells. Ten constructs were cloned for the respective variants, one of which contained two sequence alterations, one each in exons 25 and 26.

Comparison of RT-PCR amplicons from control (wild type) versus mutant transcripts revealed a full or complete effect on splicing for eight of the variants (4 in-frame exon skipping events and four frameshifts). The remaining three were showing an incomplete effect on splicing, since properly spliced transcripts were also present in addition to aberrant splice products. On protein level, three variants caused an in-frame deletion in the amino acid sequence while all the others introduced frameshifts according to in silico translation prediction. 

The study has two obvious limitations: (i) The constructs containing eleven variants of interest are representing an artificial environment for the splicing machinery because flanking intronic sequences are incomplete or only partially present and (ii) HEK293T cells are not the natural site of expression for USH2A gene transcripts. However, the authors are well aware of these limitations and discuss them appropriately. Moreover, mini gene assays are well established and frequently used in published literature as an initial tool for characterization of potential splice site effects in particular when the proper cell type or tissue is not available.

Additionally, the results of the functional assays clearly indicate an effect of the characterized variants on splicing of the pre-mRNA of the gene with deleterious consequences for most of the predicted proteins (pathogenic or likely pathogenic according to ACMG criteria). 

As discussed by the authors, additional experimental work, including expression studies in photoreceptor cells and/or retinal organoids, might be needed in order to provide conclusive evidence for a deleterious effect of the NCSSVs. Despite of this, the presented data provide important and novel information about functional consequences of sequence variations in NCSSs.

Minor comment:

The authors may wish to provide the clinical diagnoses of the respective cases, which can be retrieved from the referenced publications. All are either RP or Usher cases?

Author Response

We thank the reviewer for the positive and valuable feedback on our manuscript, ijms-1923533, entitled, “Minigene-based splice assays reveal the effect of non-canonical splice site variants in USH2A”. Below you can find our responses to the suggestions raised.

Reviewer 1:

The study has two obvious limitations: (i) The constructs containing eleven variants of interest are representing an artificial environment for the splicing machinery because flanking intronic sequences are incomplete or only partially present and (ii) HEK293T cells are not the natural site of expression for USH2A gene transcripts. However, the authors are well aware of these limitations and discuss them appropriately. Moreover, mini gene assays are well established and frequently used in published literature as an initial tool for characterization of potential splice site effects in particular when the proper cell type or tissue is not available.

Additionally, the results of the functional assays clearly indicate an effect of the characterized variants on splicing of the pre-mRNA of the gene with deleterious consequences for most of the predicted proteins (pathogenic or likely pathogenic according to ACMG criteria). 

As discussed by the authors, additional experimental work, including expression studies in photoreceptor cells and/or retinal organoids, might be needed in order to provide conclusive evidence for a deleterious effect of the NCSSVs. Despite of this, the presented data provide important and novel information about functional consequences of sequence variations in NCSSs.

Minor comment:

The authors may wish to provide the clinical diagnoses of the respective cases, which can be retrieved from the referenced publications. All are either RP or Usher cases?”

We agree with the suggestion of reviewer 1 to provide the clinical diagnoses of the respective cases. A column with the phenotype of the respective cases has therefore been added to Table 1.

Reviewer 2 Report

This manuscript, entitled “Minigene-based splice assays reveal the effect of non-canonical splice site variants in USH2A” by Reurink et al and colleagues, validates an assay to characterise the pre-mRNA splicing in Autosomal Recessive Retinitis Pigmentosa (arRP) and Usher syndrome Type 2 (US2A) which are known to be involved in disease pathology, this has a potential future therapeutic advantage in genetic diagnosis of the diseases above. The manuscript has novelty in the approach given that not all USH2A NCSSs have been functionally verified.

There are two main questions that need clarifying before recommending the manuscript for publication. First, why choosing HEK193 cells to design a splice assay for retinal disease, why not retinal cells (RPE for example). The second is why the minigenes are cloned into a RHO expressing plasmid (Rho exon 3 and 5), I found that confusing, can the author clarify?

Here are minor edits in the manuscript:

·      Any of the 11 NCSS chosen are from the 13 that already known to have splice defect? The authors should clarify this point.

·      No n= number mentioned in Figure 2 legend. How many wells transfected for each condition (WT vs M)?

·      Figure 2G, the gel shows one band in the mutant column but the schematic shows two transcripts detected (-62 and -199). Can you clarify?

·      Figure 2F, there is a faint band in the WT column same size as the mutant (~600bp) can the author comment on that?

·      A typo in first line of discussion “FigurAll”.

Author Response

We thank the reviewer for the positive and valuable feedback on our manuscript, ijms-1923533, entitled, “Minigene-based splice assays reveal the effect of non-canonical splice site variants in USH2A”. Below you can find our responses to the suggestions raised.

Reviewer 2:

This manuscript, entitled “Minigene-based splice assays reveal the effect of non-canonical splice site variants in USH2A” by Reurink et al and colleagues, validates an assay to characterise the pre-mRNA splicing in Autosomal Recessive Retinitis Pigmentosa (arRP) and Usher syndrome Type 2 (US2A) which are known to be involved in disease pathology, this has a potential future therapeutic advantage in genetic diagnosis of the diseases above. The manuscript has novelty in the approach given that not all USH2A NCSSs have been functionally verified.

There are two main questions that need clarifying before recommending the manuscript for publication. First, why choosing HEK193 cells to design a splice assay for retinal disease, why not retinal cells (RPE for example). The second is why the minigenes are cloned into a RHO expressing plasmid (Rho exon 3 and 5), I found that confusing, can the author clarify?

We understand the confusion of the reviewer, however, minigene splice assays using the pCI-neo mammalian splice vector (with RHO exons 3 and 5) and HEK293T cells are an established model to observe deviations in pre-mRNA splicing. Several studies have been published previously that employ minigene splice assays with the pCI-neo vector and/or HEK293T cells to observe potential splicing defects for retinal disease (for example PMID 20497194, 34996991, 33737949 and 29162642).

We chose this minigene splice assay model system as RHO and USH2A are not endogenously expressed in HEK293T cells, which excludes the possibility of amplifying native mRNA. Any product resulting from an RT-PCR with primers specific for RHO or USH2A will therefore be the result of our transfection. In addition, hTERT RPE-1 are retinal pigment epithelial cells. Although these are more closely related to photoreceptor cells, which is the cell type that is primarily affected in retinitis pigmentosa and Usher syndrome, we do not know how well these cells recapitulate the splicing pattern in photoreceptor cells.

Here are minor edits in the manuscript: 

  • Any of the 11 NCSS chosen are from the 13 that already known to have splice defect? The authors should clarify this point.

We understand the question of the reviewer as this is not explicitly stated in the manuscript. None of the 11 variants that we tested are part of the 13 variants that were already published to have an effect on pre-mRNA splicing. We therefore added a clarification in the last paragraph of the introduction to explain that the 11 variants that were tested in this manuscript have not been functionally validated previously.

  • No n= number mentioned in Figure 2 legend. How many wells transfected for each condition (WT vs M)?

We added a clarification in the materials and methods, in the ‘minigene splice assays’ section that each variant was tested twice in two separate biological replicates.

  • Figure 2G, the gel shows one band in the mutant column but the schematic shows two transcripts detected (-62 and -199). Can you clarify?

We agree with reviewer 2 that the band for skipping part of exon 55 (−62 nt) is barely visible on gel. However, the product of the partial exon skipping has been observed in our analysis and we believe that the effect should therefore not be neglected in this manuscript. To clarify that the band observed for the partial skipping of exon 55 was very faint, we added a sentence in the results in the “Assessment of pre-mRNA splicing using minigene splice assays” section.

  • Figure 2F, there is a faint band in the WT column same size as the mutant (~600bp) can the author comment on that?

We agree with reviewer 2 that a very faint band is visible around 600 nt. This band could not be sequenced as there was not enough material, but it may well be a similar exon skipping event as was observed for transfection with the mutant construct. We therefore added a sentence to the discussion section, following the observation of skipping of exons 25 and 26 after transfection with another wildtype construct, to elaborate on the presence of this band upon transfection of the wildtype construct for variant c.10182G>A.

  • A typo in first line of discussion “FigurAll”.

We thank the reviewer for addressing this. The typo has been removed.

Round 2

Reviewer 2 Report

The authors kindly answered all my questions and I therefore recommend the manuscript for publication.